# Relationship between Dynamic and Isometric Strength, Power, Speed, and Average Propulsive Speed of Recreational Athletes

**DOI:** 10.3390/jfmk7040079

**Published:** 2022-09-30

**Authors:** Jairo Alejandro Ferná Ortega, Dario Mendoza Romero, Hugo Sarmento, Laura Prieto Mondragón, Jeansy Alonso Rodríguez Buitrago

**Affiliations:** 1Facultad de Ciencias de la Salud, Universidad de Ciencias Aplicadas y Ambientales, Street 222 #55-17, Bogotá 668470, Colombia; 2Laboratorio de Fisiología del Ejercicio, Facultad de Educación Física, Universidad Pedagógica Nacional, Street 72 #11-86, Bogotá 110321, Colombia; 3Facultad de Ciencias de la Salud y del Deporte, Fundación Universitaria del Área Andina, Street 69 #15-40, Bogotá 110211, Colombia; 4Research Unit for Sport and Physical Activity (CIDAF), Faculty of Sport Sciences and Physical Education, University of Coimbra, 3000-456 Coimbra, Portugal

**Keywords:** stretch, dynamic and isometric, peak power, mean propulsive velocity

## Abstract

The purpose of this study was to examine the type of relationship between measures of maximal force (dynamic and isometric), maximal power, and mean propulsive velocity. In total, 355 recreational athletes, 96 women (age 20.5 ± 2.5 years; height 158.2 ± 17.3 cm; weight 61.8 ± 48.4 kg) and 259 men (age 21.0 ± 2.6 years; height 170.5 ± 12.6 cm; weight 65.9 ± 9.2 kg) were evaluated in three sessions separated by 72 h each in isometric midthigh pull exercise (ISOS) (kg), bench press maximum strength (1RM MSBP) (kg), jump height (CMJ) (m), and maximum pedaling power (WT) the maximum squat strength (1RM MSS) (kg), the mean propulsive velocity in the bench press (MPVBP) (m·s^−1^), and the peak power (PPBP) (w), mean propulsive squat velocity (MPVS) (m·s^−1^), peak power (PP) (w), maximum handgrip force (ISOHG) (kg), and 30 m movement speed (V30) (s). Significant correlations (*p* ≤ 0.01) were identified between 95% of the various manifestations of force, and only 5% presented a significance of *p* ≤ 0.05; however, when the magnitude of these correlations is observed, there is great heterogeneity. In this sense, the dynamic strength tests present the best correlations with the other strength and power tests used in the present study, followed by PPBP and PP. The results of this study complement what is reported in the literature regarding the correlation between different types of force manifestations being heterogeneous and contradictory.

## 1. Introduction

Maximum strength and power are highly relevant factors in a variety of sports and activities of everyday life. Both in performance sports and in the field of health, the relationship between strength and sports performance or muscle fitness is of great importance for researchers, trainers, and fitness professionals. This has led to the development of a wide variety of tests for the evaluation of the muscular strength of different body segments that can be grouped into isometric and dynamic and present high levels of reliability [1].

Previous research has identified correlations between isometric strength and dynamic strength or between different dynamic strength assessment tests. Likewise, relationships between 1 repetition maximum (1RM) and maximum power have been established, but the exact relationship between these two is not yet clear. The highest peak power (PP) values have been found to occur at approximately 30% of the maximal isometric force in isolated movements and at approximately 30–50% of 1RM [2].

For example, it has been identified that isometric squat strength (ISOS) correlates well with maximum squat strength (MSS) tests [3,4] and with counter-movement jump (CMJ) in high-performance athletes [5,6,7]. Relationships between CMJ and strength have also been identified: isometric and dynamic force [5,7,8], or power, strength, and displacement velocity (30), or 1RM force, CMJ, and velocity [9,10]. Other studies indicate that 1RM has a high incidence in peak power [11], and the variation between power and maximum force is approximately 50% [12], and others [13,14] observed that when adding additional resistance to a movement, the relationship between maximum force, power, and velocity tended to increase with the additional resistance.

More recently, Moss et al. [11] investigated the relationship between 1RM and PP in various percentages of 1RM using elbow flexion and found strong correlations between 1RM and peak power, PP (*r* = 0.93).

This indicates the divergence in the results between the various previous studies on the relationship between the various manifestations of force and their levels. These differences may be due to the fact that the level of correlation between manifestations of strength changes depending on a number of factors, such as training time, the athlete’s training level, gender, age, training season, type of muscles involved in the movements, the methodology used in the evaluation, the materials or the size of the samples, among others.

The purpose of the present study was to verify the possible correlations and their level between different strength evaluation methods (MSBP, MSS, ISOHG, ISOS, CMJ, V30, WT, PPBP, PP, MPVBP, and MPVS), reducing bias factors such as training time, athlete’s training level, age, training season, methodology used in the evaluation, materials, or sample size. A large sample made up of young recreational athletes of similar age and performance levels was used, who were evaluated in the different tests with standardized methodology and materials. To our knowledge, this is the first study to examine a wide range of strength assessment methods in upper and lower limbs with one of the highest numbers in the population.

For the development of the study, the following hypotheses were proposed: the relationships between maximum dynamic strength, isometric strength, and power would be very heterogeneous due to kinematic, neuromuscular, and joint factors. The second hypothesis is that the dynamic strength tests are the tests that present the strongest correlations with the rest of the strength and power measures. The third hypothesis is that handgrip strength is not a good predictor of a subject’s global maximal strength.

## 2. Materials and Methods

This study describes the relationships between the 1RM, PP, and MPV (50%, 60%, and 70% 1RM) of the dynamic tests of multiple joints (squat and bench press), as well as CMJ (height), PP in pedaling, isometric hand grip and squat strength, and displacement velocity in 30 m in recreational athletes.

### 2.1. Subjects

Three hundred thirty-five recreational athletes took part in this study. They had similar levels of training in physical activities, recreational sports (6 h/week), and no experience in strength training. A total of 96 women (age 20.5 ± 2.5 years; height 158.2 ± 17.3 cm; weight 61.8 ± 48.4 kg) and 259 men (age 21, 0 ± 2.6 years; height 170.5 ± 12.6 cm; weight 65.9 ± 9.2 kg) participated in the study, which was designed taking into account the deontological standards established in the Declaration of Helsinki; all signed informed consent and participated in six adaptation sessions and three evaluation sessions. The project was developed between 2018 and2019 and was conducted by groups of 10 subjects. During the two weeks of adaptation, exercises were carried out in each of the tests in which the participants would be evaluated with the purpose that they knew and learned the execution techniques. All the participants underwent two weeks of adaptation with four sessions per week, performing four series of ten repetitions of each of the strength evaluation tests, with the purpose of standardizing the learning of the methodologies and protocols of each test.

The study was approved by the research ethics committee of the University of Applied and Environmental Sciences.

In the first session, isometric midthigh pull exercise (ISOS) (kg), bench press maximum strength (1RM MSBP) (kg), jump height (CMJ) (m), and maximum pedaling power (WT) were assessed. In the second session, the maximum squat strength (1RM MSS) (kg), the mean propulsive velocity in the bench press (MPVBP) (m·s^−1^), and the peak power (PPBP)(w) were assessed. In the third session, mean propulsive squat velocity (MPVS) (m·s^−1^), peak power (PPS) (w), maximum hand grip force (ISOHG) (kg), and 30 m movement speed (V30) (s) were assessed.

Between each of the sessions, there was a 72-hour recovery period where the participants performed their daily activities and refrained from vigorous activities. The tests were always performed at the same time (2–4 p.m.) to avoid the effects that circadian rhythms have on neuromuscular performance. Before evaluating each test, a general warm-up was performed with a total duration of 10 min distributed as follows: 5 min of band jogging at a velocity of 8 km/h and 5 min of active stretching and joint mobility.

### 2.2. Procedures

#### 2.2.1. Evaluation of the Maximum Dynamic Force

The 1RM was evaluated in the bench press and squat according to the protocol of Sanchez et al. [15]. A specific warm-up was carried out using only the weight of the bar; 3 series of 8 repetitions were carried out, and after three minutes of recovery, the 1RM was estimated. A Smith machine was used, which allows the vertical movement of the bar in a certain path, with a very low friction force between the bar and the support rails. The Smith machine did not have any type of counterweight mechanism, acting identical to free weights. For the recording of control of the velocity of movement of the bar, a linear velocity transducer (T-FORCE Dynamic Measurement System2, Ergotech Consulting SL, Murcia, Spain) was used, which provided auditory and visual feedback in real-time with a sampling rate of 1000 Hz, which automatically determined the eccentric and concentric phases of each repetition, as well as the propulsive phase of the concentric phase during which the acceleration of the bar is greater than the acceleration due to gravity [16,17].

The bench press test starts with a load of 10 kg for women and 20 kg for men, with which four repetitions are performed, progressive increases of 3–5 kg are made, and with them, 3 repetitions are executed with each weight until the achieved MPV was less than 0.50 m·s^−1^ [16]. From that moment on, the increases were from 1 to 2 kg, and two repetitions were performed with each weight, even when the participants were unable to perform the 180° extension and the MPV was less than or equal to 0.20 m·s^−1^ [18]. The last load that each subject was able to perform correctly up to full extension was considered his 1RM. The test is performed in a supine position with the feet resting on the bench, the hands placed on the bar slightly more open than the width of the shoulders (5–7 cm). The amplitude of the grip was measured so that it could be reproduced in each series. The participants were instructed to lower the bar in a slow and controlled manner until they reached one centimeter from the top of the xiphoid process and wait until they heard the order of the evaluator to perform the extension of the arms at maximum velocity without raising the trunk and the shoulders of the bench. The pause lasted for approximately 2 s [19] in order to avoid the rebound effect and allow more reproducible and consistent measurements. The breaks between series were three minutes for loads less than 80% of the estimated 1RM and 5 min for loads greater than 80% of the estimated 1RM [20]. During the execution of the tests, the participants were motivated by the evaluators to do their best.

The full squat force was performed using the same protocol as described above, but started with a load of 20 kg for women, 30 kg for men, and 10 kg increments until the achieved MPVMPV was less than 0.60 m·s^−1^; the increments were 3–5 kg. Subjects begin in an upright position with knees and hips fully extended, feet shoulder-width apart, and bar supported at acromion level. This position was carefully checked so that it could be reproduced in each series. For reasons of standardization and safety, the participants descend in a controlled manner at an average velocity of ~0.50–0.60 m·s^−1^, until they reach a flexion that leads to a tibiofemoral angle of 35–40° in the sagittal plane, measured with a goniometer (Nexgen Ergonomics, Point Claire, QC, Canada) to achieve a deep squat [21]. In this position, there was a pause of 2 s, and at the order of the evaluator, they carried out an extension at maximum velocity.

#### 2.2.2. Evaluation of MVP-MPV-MPV PPS and PPBP

With the results obtained in the 1RM test, in bench press and squat, the participants had to execute at 50%, 60%, 70%, and 80% of 1RM, two repetitions in each movement on the Schmith machine, under the protocols described above for each test. A specific warm-up was performed using only the weight of the bar, and three sets of eight repetitions were performed.

### 2.3. Vertical Jump

After a 5-minute warm-up on the cycle ergometer, the subjects performed a counter-movement jump (CMJ). The height of the jump was calculated using an infrared timer system (Optojump, Microgate^®^,Bolzano Italia (precision of 1/1000 s), which, through the time spent in the displacement of the center of gravity during the flight phase, estimates the jump height (h) as follows: h = (gx ft2), where g represents the acceleration of gravity (9.81 m·s^−2^) [22]. How the take-off and landing position can affect the jump flight, all participants were instructed to keep their legs extended during the flight time. They started from a standing position and performed a downward knee flexion movement until approaching a 90° angle and immediately a push at maximum velocity, always keeping the hands on the hips [23]. Participants received feedback on the results obtained in each of their attempts. Five attempts were made, separated by 3 min of recovery between each one; highest and the lowest value were discarded, and the average was made with the remainder that was used for the analysis [24].

### 2.4. Maximum Pedaling Power

It was evaluated on a Monark 834 E brand cycle ergometer (Monark exercise, Varberg, Sweden), adjusting the saddle to the height of the iliac spine and with a load equivalent to 6.7% of body weight [25]. The participants had to pedal at the maximum possible velocity and remain seated without getting up from the chair. A specific five-minute warm-up was performed on the cycle ergometer with a pedaling frequency of 40 RPM and a resistance of 2% of body weight, and they performed 5-second sprints at the end of each minute. After three minutes of rest, the test was carried out [26].

### 2.5. Sprint 30 m

Two accelerations were made over a distance of 30 m on an athletics track, and the time was recorded using a system of infrared light photocells model WL34-R240 (Sick^®^ Düsseldorf, Germany), which were located at 0 and 30 m [27]. Five minutes of recovery were allowed between each execution, and for analysis purposes, the best record was taken. The output was high, with the starting foot placed behind the first photocell; the recording started when the participant crossed the infrared light beam and stopped when the infrared lights of the second photocell were crossed. A specific warm-up was carried out where the acceleration of 10 m, 15 m, and 20 m was carried out. In the end, the participants walked slowly back to the starting line. After three minutes of recovery, the test was carried out.

### 2.6. Isometric Force

The grip strength of both hands was assessed with the dynamometry method using a Takei dynamometer (Scientific Instruments Co., Ltd., Tokyo, Japan). Two attempts were made with each hand with 3 min recovery periods, and the best result was recorded. The lower limbs were assessed using the Takei 5002 dynamometer (Scientific Instruments Co., Ltd., Tokyo, Japan) in a squat pull. Participants performed a five-minute warm-up on a cycle ergometer at 70 RPM. After 3 min of recovery, they were placed on the platform of the dynamometer, adjusting the height of the grip to achieve flexion of the knees at an angle of 90 degrees, measured with a goniometer. The participants performed the greatest force, trying to extend the knees and keeping the back straight in order to focus the effort on the quadriceps muscles. Each one made three attempts with a duration of 3 s, the recovery time between each attempt was 3 min, and the best of the three attempts was taken.

### 2.7. Statistical Analysis

The data of the evaluated variables are summarized in means and standard deviations. They are presented in tables comparing men with women by means of a test for independent samples, T-Student. Pearson correlations were performed to assess the strength of the relationship between the variables in the two groups with a significance level of *p* ≤ 0.05. The correlations found as statistically significant in this research had an average statistical power of 0.83. The statistical package IBM SPSS version 26 (Universidad Santo Tomás license) was used for the inferential tests, and the correlations were plotted with the software. Power calculations for correlations were calculated using G * Power (Version 3.1, University of Dusseldorf, Germany). A qualitative rating scale was used according to the magnitude of the observed correlation. Weak for values less than 0.40; moderate for values between 0.41 and 0.60; strong for values between 0.61 and 0.80, and very strong for values between 0.81 and 1.

## 3. Results

Table 1 shows the results obtained in each of the groups in the various strength assessments that were made. As expected, a significant difference is observed between men and women in all tests.

Table 2 shows how each of the tests correlates with the others, with how many is it correlated, what is the level of significance, and the magnitude of these correlations. In Figure 1, the correlations between the measures of absolute maximum strength, power, and MPVP are presented in the total population; in Figure 2, these same correlations in the population of women; and in Figure 3, that of men. In Figure 4, the correlations between the power measurements and MPV are presented.

A significant correlation was identified between the various manifestations of force evaluated (Figure 1 and Figure 2). However, when the magnitude of these correlations is observed, there is great heterogeneity. In this sense, the MSBP is the test that presents the best correlations with the other strength and power tests used in the present study. I present strong levels of correlation (*r* = from 0.80 to 0.81) with 67% of the tests (CMJ, ISOHG, WT, MSS, V30, PPS, and PPBP). It is followed by the MSS, which presented the same type of correlations with 58% of the tests (CMJ, ISOHG, WT, ISOP, MSBP, PPBP, and PPS). The ISOS, PPS, and PPBP tests presented strong levels of correlation with 50% of the variables. Strength ISOP with PPS, PPBP, FMP, MSS, WT, ISOM, PPS with PPBP, FMP, ISOP, MSS, WT, CMJ and PPBP with MSS, ISOP, MSS, WT, and ISOM. The CMJ, ISOM, WT, and MSS did so with 31% of the variables, while the set of velocity tests (VM30, MPVS, and MPVBP) correlated with 17% and 8%, respectively.

When the population is disaggregated according to sex (Figure 2, Figure 3 and Figure 4, the same behavior occurs; the MSBP and the MSS continue to be the tests that present the best results. In the group of women, it is significantly correlated (*p* < 0.01, *p* < 0.05) with the various strength tests evaluated, except with the MPVBP70, whose correlation is not significant. However, only 22% of these correlations are strong for MSBP and 18% for MSS. On average, 74% of the correlations between the other tests show low levels of correlation. In the group of men, the MSS and the MSBP are significantly correlated (*p* < 0.001, *p* < 0.05) with all the manifestations of force except with MPVS and MPVBP, and the magnitude of these correlations is only strong (from 0.80 to 0.81) with the 25% of the tests. On average, 71% of the correlations between the other tests show low levels of correlation.

## 4. Discussion

The results of the present study reaffirm the hypothesis that would indicate that the dynamic strength tests MSBP and the MSS are the strength evaluation tests that best predict the different manifestations of force. The MSBP and MSS are the strength evaluation tests that present the highest levels of correlation with most other strength evaluation tests, making them the most accurate indicators of the overall strength of a subject. Likewise, it was confirmed that despite the existence of significant correlations between the manifestations of force, the levels of these correlations are very heterogeneous and that handgrip strength is not a good indicator of the overall strength of a subject.

The difference in correlation in the measurements of strength, power, and speed between the various studies may be due to factors associated with the study population, such as gender, level of training, time in the training, and the time of the season.

To the best of our knowledge, this study is the first one that integrates various manifestations of strength in both upper and lower limbs in populations of recreational athletes, and due to the high number of participants, it has high statistical power for establishing correlations. Likewise, given the conditions of being recreational athletes not specialized in a sport, they had the same conditions to develop the different types of tests and events that in several of the reviewed studies were reported as limitations.

### 4.1. Maximum Strength in Bench Press

The bench press exercise is receiving increasing interest as a field test and is considered the gold standard for quantifying upper body muscle strength. However, to our knowledge, there are few studies that identify the relationship of this measure of strength with measures of the lower limbs. Of this low number of studies, most are focused on the relationship with ISOHG.

In the present study, a very strong correlation was identified between MSBP and MSS, which was also observed in previous studies (Mc Guigan (*r* = 0.96; *p* < 0.05) [3]; Ferland et al. [28] (*r* = 0.77; *p* < 0.05) in weightlifters and soccer players.

MSBP has been correlated with other upper limb strength modalities such as push-ups or ISOHG. In the present study, a moderate correlation was observed between MSBP and ISOHG, which confirms what has been reported in previous studies carried out in children and women who are breast cancer survivors [29,30] that defined ISOHG as a poor indicator of MSBP, and therefore, these two modalities, which are used to quantify muscle strength in the upper extremities, are not interchangeable.

Several studies have investigated the relationship between MSBP and ISOHG, and a relationship between both tests has been identified; however, these relationships are quite scattered (*r* = from 0.570 to 0.78; *p* < 0.05). This variation may be affected by the different joint angles used during the ISOHG in the various studies and differences in the biomechanics of exercise. MSBP is an exercise that involves several joints and requires the simultaneous recruitment of multiple large muscle groups, which contract in eccentric and concentric actions. In contrast, the ISOHG test is an exercise that involves a single joint with few muscles and small size. Additionally, it has been indicated that grip strength is not a good indicator of identifying changes in strength and muscle structure after a training program [31].

Likewise, strong correlations were identified between MSBP and ISOS and CMJ that had also been reported in a previous study [3] (*r* = 0.99; *p* < 0.05). Additionally, in the present study, strong correlations were found of the MSBP with the WT and PPBP and very strong with PPS, which could not be verified because no previous studies were identified that inquired about these associations.

Contrary to what was stated in the systematic review by Lum [32], although the 1RM bench press test does not provide data on the force-time characteristics for a better understanding of an athlete’s force-generating capacities, in the present study, it was the one that presented better correlations with both the maximum strength and power and speed tests.

### 4.2. Maximum Strength in Full Squat

According to the hypotheses of this study, the MSS was the second test that presented the highest magnitudes of correlation with the largest number of variables and, as mentioned above, with a strong correlation between them. The results of the present study report strong correlations between MSS, the CMJ, and the VM30, which were observed in previous studies with *r* = 0.78, *p* < 0.001 with the CMJ, and *r* = −0.71, *p* < 0.001 with time in the 30 m test [2,3,9,33,34,35].

Therefore, lower body musculature strength appears to play a role in maximal running speed, CMJ, and pedaling power. In this regard, Confort et al. [33] indicate that MSS largely determines performance in jump height and running velocity in elite and youth soccer players. These correlational findings suggest that jumping power can be increased with the improvement of the 1RM squat, although there are other factors that may contribute to CMJ performance, such as (a) a greater active state of the muscle as a result of the anterior eccentric phase, (b) storage and use of elastic energy, and (c) a stretch (mitotic) reflex that occurs as a result of eccentric action. In the present study, the strong correlation level of CMJ with maximum squat strength was presented at 1RM and was weak, with values of 50–70%, different from those observed by Stone et al. [2] that identified in a population of subjects with high experience in squatting, better correlations at 50% of 1RM. One of the reasons that could explain this difference is the characteristics of the participants who in the present study had no experience in squat strength training. To our knowledge, there is little evidence reporting non-significant correlations between relative squat 1RM and CMJ height.

Regarding the correlation between MSS and ISOS, in the present study, strong correlations were observed, which have been reported in several previous studies, but with great variability in magnitude. For example, McGuigan [3] observed an almost perfect correlation between the measures of ISOP and MSS. This controversy may be due to the difference in joint angles used when performing ISOS. ISOS performed at a 90° knee angle demonstrated a higher correlation with the 1RM full squat compared to that performed at a 120° knee angle (*r* = 0.864 vs. 0.597).

Therefore, the variation in the magnitude of the relationship between ISOS and MSS observed in several studies could be related to the use of different knee and hip angles during isometric tests [1,3,5,36,37].

However, it is important to note that despite the evidence of this correlation, the validity of ISO tests to predict or correlate with performance in human activity, which is dynamic, has previously been questioned since neural activation and length of muscles are very different in ISO actions and dynamics. Differences in muscle length in a concentric contraction can result in an afferent signal that is different from that stimulated in an ISO contraction. The difference in afferent signaling can potentially result in a greater efferent command in a concentric contraction [38,39,40]. More importantly, the changes in isometric and dynamic strength consequent to a dynamic strength training program are unrelated. Therefore, the mechanisms that contribute to improving dynamic strength appear to be unrelated to the mechanisms that contribute to improving isometric strength [40].

Likewise, in the present study, strong correlations were observed between MSS and ISOHG and PPBP and very strong with PPS, which could not be verified because no previous studies were identified that inquired about these associations.

### 4.3. Isometric Strength

Isometric strength tests are relatively simple to administer, present minimal risk of injury, have high test–retest reliability, can detect subtle changes in strength, and are considered less strenuous than the 1RM test [34].

The isometric squat has been used to detect changes in kinetic variables because of training; however, there is controversy in its application to dynamic multi-joint tasks.

Studies examining the relationship between IsoTests and maximum velocity in a sprint reported significant relationships between the force-time characteristics of ISOP and sprint acceleration performance in different athletes [41,42]. Moderate correlations were observed in the present study between VM30 and ISOP, such as those reported by several studies such as Townsend et al. [37], who reported that the average velocity and the maximum velocity reached during a 20 m sprint were significantly correlated with ISOP (*r* = 0.704; *p* < 0.01 and *r* = 0.536; *p* < 0.01, respectively). Thomas et al. [41] reported strong correlations between ISOP and the sprint times of 5 and 20 m in college football and rugby players.

It is important to note that most of the studies that compare the relationship between ISOP and running velocity have been observed in the acceleration phase of the sprint, and very little has been investigated about the relationship between the force-time characteristics in the IsoTests and the maximum velocity of the sprint. For this reason, the present study addresses the analysis of the relationship between the maximum force in ISOP and the maximum displacement velocity over a longer distance, as suggested by the systematic review published by Lum et al. [32].

When comparing the ISOP with the CMJ in the present study, a moderate correlation was observed, which is within the range of that reported in other studies, *r* = from 0.346 to 0.820; *p* < 0.05 [3,32,43,44].

However, when carefully observing the data reported by the literature against this correlation, great variability in the relationships is observed [5,44,45]. Other studies, on the contrary, did not identify significant correlations [5,38].

The discrepancy in the magnitude of the correlations between the jump height achieved during the CMJ and the maximum ISOS force between the studies could be due to the difference in the angle of the knee joint where the force begins in both exercises [46], which differs between 90° to 135°, or the velocity and depth with which the CMJ is executed [45]. It has been shown that isometric movement position can strongly influence the relationships that are observed with dynamic tasks ([3]).

On the other hand, in the present study, strong correlations of ISOP with ISOHG, WT, PPBP, and PPS were observed, which could not be verified because no previous studies were identified that inquired about these associations.

### 4.4. Hand Grip Strength

Hand grip strength has become in the clinical field an indicator of physical fitness and physical health, as well as an indicator of risk for various chronic diseases. However, the ISOHG, as observed in the present study, is not the best indicator of the strength and maximum power of an individual.

In the present investigation, as in previous investigations, moderate correlations of ISOHG MSBP were identified, which were previously described. Additionally, moderate correlations of ISOHG with CMJ, WT, MSBP, MSS, PPBP, and PPS) were found, which could not be verified because there were no previous investigations of these associations.

### 4.5. Anaerobic Power Tests

Anaerobic power tests include strength–velocity tests, vertical jump tests, stair tests, and cycle ergometer tests. The maximum anaerobic power values obtained with these different protocols, despite evaluating different aspects of power and anaerobic capacity, are well correlated but can vary according to sex, type of sport practiced by the subject, and age.

### 4.6. CMJ

The ability to jump is an important skill required for good performance in many sports, and it has often been used to assess lower extremity power and jumping ability. In the present investigation, as in previous investigations, correlations were identified in different types of CMJ populations with ISOS, MSBP, and MSS, which were previously described.

Several studies have compared jumping and lower limb isometric strength with the purpose of identifying how one determines the other, and significant relationships have been reported (*r* = from 0.346 to 0.820; *p* > 0.05) [1,5,7,37,41,43].

Additionally, it was observed a strong correlation of the CMJ with the VM30 test that was reported in previous studies in populations of recreational athletes at distances of 20 m [34,35,47] and 30 moderate with the WT [36,39,48].

In the present study, moderate correlations of CMJ with ISOHG and PPBP and strong with PPS were observed, which could not be verified because no previous studies were identified that inquired about these associations.

### 4.7. Maximum Anaerobic Power Wingate Test (WT)

The Wingate test has been used in different sports to determine the maximum anaerobic power of a subject. In the present investigation, as in previous investigations, correlations were identified in different types of populations of WT with ISOP, MSBP, MSS, and CMJ, which were previously described.

Regarding the correlation of WT with velocity tests, various studies have identified significant correlations in various sports, such as with velocity tests in basketball [49] or 1500 m in skating [50]. The results of this study confirm what was stated in previous studies, which indicate that the power obtained by the WT test is a reliable predictor of running velocity. Likewise, the CMJ and the SJ were correlated with the peak power and average power of the WT in different athletes [36].

Additionally, in the present study, moderate correlations were observed between WT and ISOHG and strong correlations with PPBP and PPS, which did not could be verified because no previous studies were identified that inquired about these associations.

### 4.8. Velocity 30 m

Velocity is another critical skill required for successful performance in many sports. The 30 m distance is a measure of maximum velocity. In the present investigation, as in previous investigations, correlations were identified in different types of populations of VM30 with ISOS, MSBP, MSS, CMJ, and WT, which were described above.

Studies examining the relationship between IsoTests, sprint acceleration, and maximal sprint performance have reported significant relationships between IsoTests and sprint acceleration performance in different athletes [32].

Additionally, in the present study, moderate correlations of VM30 with ISOHG, PPBP, and PPS were observed, which could not be verified because no previous studies were identified that inquired about these associations.

### 4.9. Mean Propulsive Velocity (MPV)

The MVP, regardless of the 1RM percentage and the type of movement (squat and chest press), showed very weak correlations with all strength tests. The 50% MVP in chest press showed moderate correlations with the power tests (WT, PPBP). To our knowledge, this is the first study to observe the correlation of the MVP with other modalities of evaluation of maximum strength and power; therefore, the results cannot be verified.

The present study provides evidence of the strong relationship (*p* < 0.01) between the maximum strength and power tests. As has been described throughout this chapter, there is a divergence between various studies regarding the relationship between these two manifestations of force. The exact relationship between maximal strength (measured by 1RM) and power output is unclear. Stone et al. [2] indicate that maximal strength is the basic quality that affects power output. It was initially established that maximal force affects power in a hierarchical manner with decreasing influence as external load decreases, up to a point where other factors, such as rate of force development, become more important. This argument indicates that maximal force should have its greatest influence on power output at heavy loads and that light loads should be less influenced by maximal force levels. However, some evidence indicates that maximal force may have a greater influence on power output over a broader range of loads than Moss et al. [11] investigated the relationship between 1RM and PP at various percentages of 1RM using elbow flexion and identified strong correlations between 1RM and maximal PP output (*r* < 0.93), as well as a strong correlation between PP output at low % of 1RM (*r* < 0.73). In the present study, a strong correlation (*r* = 0.81) was observed between maximal strength and PP in both the squat and chest press.

## 5. Conclusions

This research examines the relationship between different forms of expression of strength and power. The results of this study indicate that although it is true that there is a correlation between the various manifestations of strength, there is great heterogeneity in the level of said correlation, and it is equally different according to gender. Reports in the literature on the correlation between different types of force are heterogeneous and contradictory. There is a large number of studies that observe strong, moderate, or weak correlations between various manifestations of strength or that do not report significant correlations.

The results of this study indicate that subjects with high levels of strength in the squat test have better speed in 30 m races, height in the jump test, and in pedaling power, aspects that are important for sports that require these qualities, although a strong correlation does not imply cause and effect. Additionally, it was observed that the chest press test is the most efficient method to diagnose the general strength of a subject and that, on the contrary, the prehensile strength is not the best indicator of the global strength of a subject.

Additional research is needed to understand and explain possible reasons for these relationships.

## Figures and Tables

**Figure 1 jfmk-07-00079-f001:**
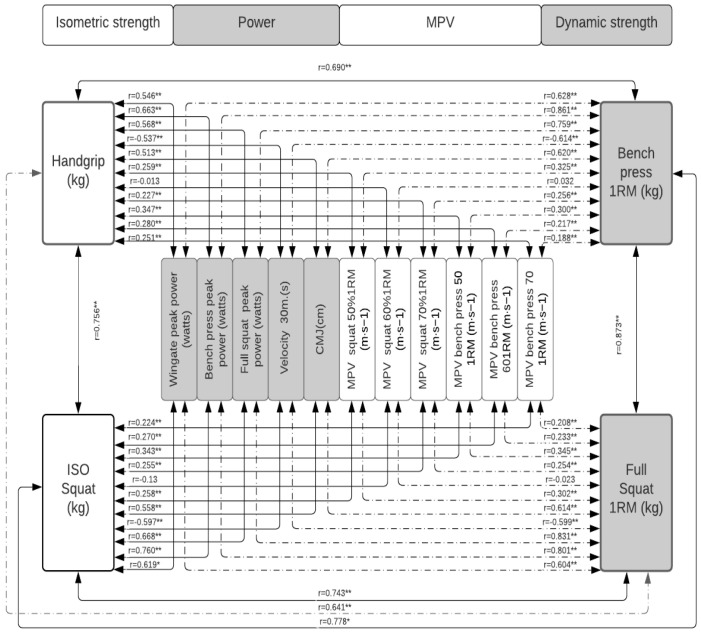
Correlation between measures of absolute maximum strength and power in the total population. Significance: * *p* < 0.05; ** *p* < 0.001.

**Figure 2 jfmk-07-00079-f002:**
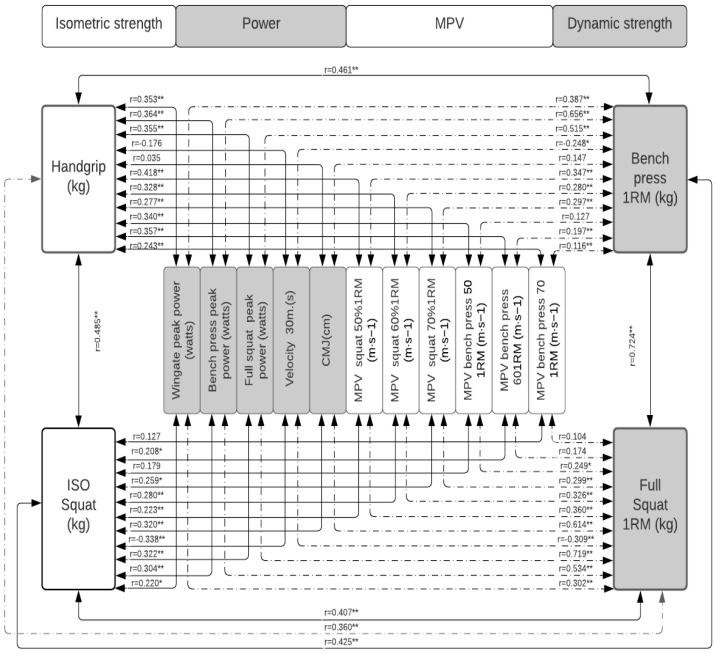
Correlation between the measures of absolute maximum strength and power in the group of women. Significance: * *p* < 0.05; ** *p* < 0.001.

**Figure 3 jfmk-07-00079-f003:**
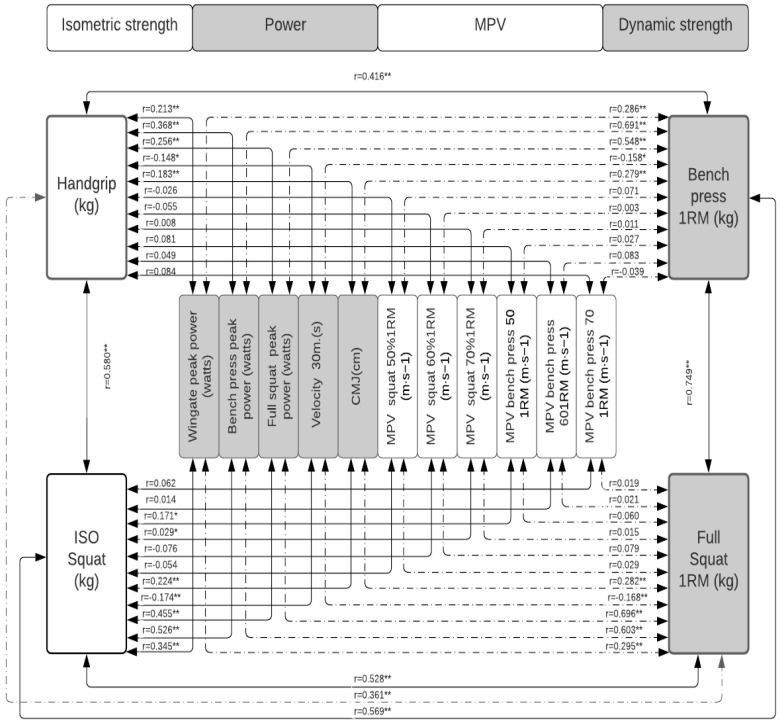
Correlation between the measures of absolute maximum strength and power in the group of men. Significance: * *p* < 0.05; ** *p* < 0.001.

**Figure 4 jfmk-07-00079-f004:**
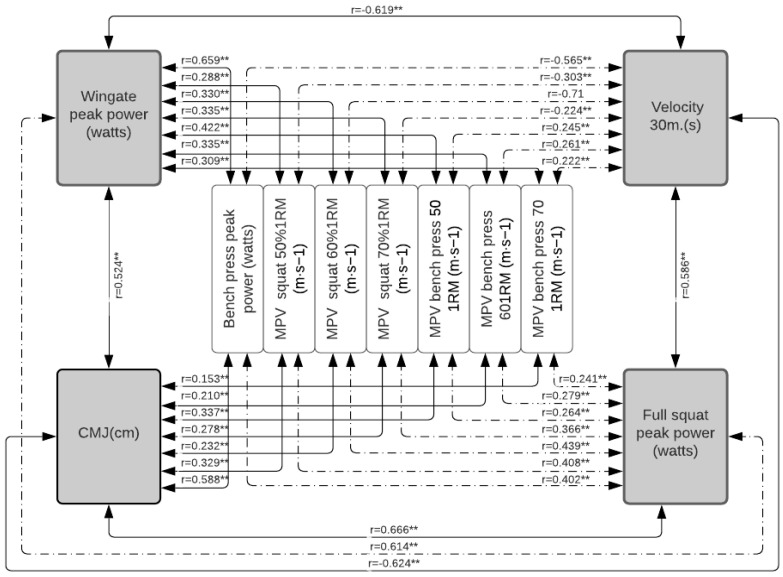
Correlation between power measurements in the total population. Significance: ** *p* < 0.001.

**Table 1 jfmk-07-00079-t001:** Values obtained (mean and SD) in the different evaluations of strength and power.

Variables	Women (*n* = 96)	CV %	Men (*n* = 259)	CV %	*p*-Value
Max hand grip force right (kg)	25.8 ± 4.3	0.15	36.7 ± 6.4	0.16	0.001
Max hand grip force left (kg)	24.4 ± 4.0	0.15	34.9 ± 6.6	0.18	0.003
Squat force max. isometric (kg)	66.9 ± 10.9	0.29	103.3 ± 18.7	0.24	0.001
Squat max force (kg)	39.4 ± 10.8	0.22	67.5 ± 12.6	0.15	0.003
Maximum bench press force (kg)	33.1 ± 8.0	0.28	59.3 ± 11.4	0.16	0.001
CMJ (cm)	24.0 ± 4.7	0.15	34.4 ± 5.3	0.13	0.003
Max velocity 30 m (s)	5.8 ± 0.4	0.17	4.4 ± 0.3	0.12	0.004
Power max. Wingate (watts)	321.1 ± 87.8	0.17	502.5 ± 102.2	0.17	0.001
Peak bench power (watts)	229.5 ± 65.0	0.27	481.5 ± 105.3	0.2	0.001
Squat peak power (watts)	413.1 ± 112.0	0.21	733.8 ± 169.6	0.11	0.001
MPVS 50% 1RM (m·s^−1^)	0.81 ± 0.09	0.25	0.87 ± 0.08	0.14	0.05
MPVS 70% 1RM (m·s^−1^)	0.55 ± 0.10	0.19	0.61 ± 0.09	0.12	0.01
MPVBP 50% 1RM (m·s^−1^)	0.82 ± 0.09	0.21	0.92 ± 0.10	0.11	0.05
MPVBP 70% 1RM (m·s^−1^)	0,51 ± 0.10	0.22	0,57 ± 0.10	0.16	0,05

**Table 2 jfmk-07-00079-t002:** Quantity, significance level, and magnitude of the correlations between the different strength tests.

Variables	Gender	Variables with Which It Correlates	%	Significance Level		Correlation Magnitude			
	W (*n* = 96), M (*n* = 259) T (*n* = 355)			*p* < 0.05	*p* < 0.01	Weak	Moderate	Strong	Very Strong
CMJ (cm)	W	5	42%	20%	80%	80%	20%	0%	0%
	M	10	83%	0%	100%	90%	10%	0%	0%
	T	12	100%	0%	100%	33%	33%	33%	0%
Max hand grip force right (kg)	W	10	83%	10%	90%	70%	30%	0%	0%
	M	8	67%	13%	88%	75%	25%	0%	0%
	T	12	100%	0%	100%	33%	33%	33%	0%
Power max. Wingate (watts)	W	8	67%	50%	50%	100%	0%	0%	0%
	M	10	83%	0%	100%	50%	25%	25%	25%
	T	12	100%	0%	100%	25%	42%	33%	0%
Squat max force (kg)	W	11	92%	9%	91%	73%	9%	18%	0%
	M	8	67%	0%	100%	50%	25%	25%	0%
	T	12	100%	0%	100%	33%	8%	42%	17%
Max velocity 30 m (s)	W	5	42%	20%	80%	60%	40%	0%	0%
	M	7	58%	43%	57%	100%	0%	0%	0%
	T	12	100%	0%	100%	33%	50%	17%	0%
Squat force max. isometric (kg)	W	10	83%	30%	70%	80%	20%	0%	0%
	M	8	67%	0%	100%	38%	63%	0%	0%
	T	12	100%	0%	100%	33%	17%	50%	0%
Maximum bench press force (kg)	W	9	75%	11%	89%	44%	33%	22%	0%
	M	8	67%	13%	88%	38%	38%	25%	0%
	T	12	100%	0%	100%	33%	0%	58%	8%
Peak bench power (watts)	W	10	83%	30%	70%	70%	20%	10%	0%
	M	11	92%	18%	82%	64%	27%	9%	0%
	T	12	100%	0%	100%	25%	25%	50%	0%
Squat peak power (watts)	W	11	92%	9%	91%	55%	36%	9%	0%
	M	11	92%	18%	82%	55%	36%	9%	0%
	T	12	100%	0%	100%	33%	17%	42%	8%
MPVBP 50% 1RM (m·s^−1^)	W	7	58%	57%	43%	86%	0%	14%	0%
	M	7	58%	14%	86%	86%	14%	14%	0%
	T	12	100%	0%	100%	67%	25%	8%	0%
MPVBP 70% 1RM (m·s^−1^)	W	6	50%	33%	67%	83%	0%	17%	17%
	M	5	42%	0%	100%	80%	20%	20%	0%
	T	12	100%	0%	100%	92%	0%	8%	0%
MPVS 50% 1RM (m·s^−1^)	W	9	75%	11%	89%	900%	75%	11%	89%
	M	6	50%	17%	83%	83%	0%	17%	0%
	T	12	100%	0%	100%	83%	8%	8%	0%
MPVS 70% 1RM (m·s^−1^)	W	9	75%	33%	67%	89%	0%	11%	0%
	M	7	58%	14%	86%	71%	0%	14%	0%
	T	12	100%	0%	100%	92%	0%	8%	0%

## Data Availability

The data that supports the results of this information is found in the databases of the exercise physiology laboratory and is not publicly available due to the data protection law.

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
