# Peer review of "Relationship between Dynamic and Isometric Strength, Power, Speed, and Average Propulsive Speed of Recreational Athletes"

_jfmk, 2022, doi:10.3390/jfmk7040079_

Round 1

Reviewer 1 Report

Thank you for the opportunity to review this manuscript entitled “Relationship between dynamic, isometric strength, power, speed and average propulsive speed, of recreational athletes”. This is one research for studying the relationship between measures of maximal force (dynamic and isometric), maximal power, and mean propulsive velocity.

The authors have conducted a laboratory study to assess the relationships between different types of strength.

I think this is good research to improve our knowledge about different types of force manifestations and them correlations.

Overall, I found some aspects should be reviewed.

I propose some minor corrections.

Line 93: full stop is needed.

Table 1: you should describe the acronym (CV)

Legend of figure 1: to check.

Line 402: It isn't appropriated to use the first person "I present...". Review this mistake.

Lines 423-425: I don’t understand the meaning of this sentence. Re-write or complete the information.

Line 470: type mistake again.

In the whole document the type writing should be reviewed.

CONCLUSIONS

You have to rewrite the conclusions. The functions of this part of the research article is to restate the main argument. It reminds the reader of the strengths of your main findings and provides a clear interpretation of the results of your research in a way that stresses the significance of your study, especially in the way of synthesizing.

Author Response

Overall, I found some aspects should be reviewed.I propose some minor corrections.

Point 1: Line 93: full stop is needed.

Response 1: The change is made

Point 2: Table 1: you should describe the acronym (CV)

Response 2:

Point 3: Legend of figure 1: to check.

Response 3: It was modified

Point 4: Line 402: It isn't appropriated to use the first person "I present...". Review this mistake.

Response 4: It was modified

Point 5: Lines 423-425: I don’t understand the meaning of this sentence. Re-write or complete the information.

Response 5: It was modified

Point 6: Line 470: type mistake again.

Response 6: It was modified

In the whole document the type writing should be reviewed.

Point 7: CONCLUSIONS: You have to rewrite the conclusions. The functions of this part of the research article is to restate the main argument. It reminds the reader of the strengths of your main findings and provides a clear interpretation of the results of your research in a way that stresses the significance of your study, especially in the way of synthesizing.

Response 7: The change is made

This research examines the relationship between different forms of expression of strength and power. The results of this study indicate that although it is true that there is a correlation between the various manifestations of strength, there is great heterogeneity in the level of said correlation and it is equally different according to gender. Reports in the literature on the correlation between different types of force are heterogeneous and contradictory. There is a large number of studies that observe strong, moderate or weak correlations between various manifestations of strength or that do not report significant correlations.

The results of this study indicate that subjects with high levels of strength in the squat test have better speed in 30-meter races, height in the jump test and in pedaling power, aspects that are important for sports where require these qualities, although a strong correlation does not imply cause and effect. Additionally, it was observed that the chest press test is the most efficient method to diagnose the general strength of a subject and that, on the contrary, the prehensile strength is not the best indicator of the global strength of a subject.

Additional research is needed to understand and explain possible reasons for these relationships.

Reviewer 2 Report

This is an interesting investigation which examines the interrelationship between maximal strength, power and dynamic athletic performance in a large and young population of people. The study is very well written and the results indicate that maximal strength of the upper and lower body is a key determinant of power and dynamic athletic performance. A Couple of minor issues that should be corrected prior to further consideration for publication. The Introduction section should be reviewed and modified to clearly highlight the deficiencies in the existing research and how the current study is of original value relative to the existing Literature. It is simply not clear of the novel value of the study . Is unclear as to the training status of the participants. There is a very large cohort which is a huge plus of this study. However there should’ve been some inclusion or exclusion criteria in the study. Details about the study participants are also relevant in terms of the reliability and validity of the strength assessments and the fact that a learning effects may have occurred between the series of tests. It is recommended, that subgroups of participants be explored based on training status The Result section also needs minor adjustments. The Figures reported occupy large space of manuscript and a highly confusing. The key results should be reported in a more concise table with results for also reported in the text. The discussion focuses mainly on the results of the upper and lower body muscle strength test. The discussion section should also encompass aspects of dynamic athlete performance and how in the relationship is variables and how this investigation contributes to our understanding of the relationship in between strength and power in athletes.

Author Response

This is an interesting investigation which examines the interrelationship between maximal strength, power and dynamic athletic performance in a large and young population of people. The study is very well written and the results indicate that maximal strength of the upper and lower body is a key determinant of power and dynamic athletic performance. A Couple of minor issues that should be corrected prior to further consideration for publication.

Point 1. The Introduction section should be reviewed and modified to clearly highlight the deficiencies in the existing research and how the current study is of original value relative to the existing Literature. It is simply not clear of the novel value of the study .

Response 1.

The following is added:

This indicates the divergence in the results between the various previous studies in the relationship between the various manifestations of force and their levels. These differences may be due to the fact that the level of correlation between manifestations of strength changes depending on a number of factors, such as training time, the athlete's training level, gender, age, training season, type of muscles involved in the movements, the methodology used in the evaluation, the materials or the size of the samples, among others.

The purpose of the present study was to verify the possible correlations and their level between different strength evaluation methods (MSBP, MSS, ISOHG, ISOS, CMJ, V30, WT, PPBP, PP, MPVBP and MPVS), reducing bias factors such as training time, athlete's training level, age, training season, methodology used in the evaluation, materials or sample size. A large sample made up of young recreational athletes with similar age and performance level was used, who were evaluated in the different tests with standardized methodology and materials. To our knowledge, this is the first study to examine a wide range of strength assessment methods, in upper and lower limbs with one of the highest numbers in the population.

For the development of the study, the following hypotheses were proposed: the relationships between maximum dynamic strength, isometric strength and power would be very heterogeneous due to kinematic, neuromuscular and joint factors. The second hypothesis is that the dynamic strength tests are the tests that present the strongest correlations with the rest of the strength and power measures. The third hypothesis is that handgrip strength is not a good predictor of a subject's global maximal strength.

Point 2. Is unclear as to the training status of the participants.

Response 2.

The following is added:

With similar levels of training in physical activities, recreational sports (6h/week) and no experience in strength training.

Point 3. There is a very large cohort which is a huge plus of this study. However there should’ve been some inclusion or exclusion criteria in the study.

Response 3. The study previously defined inclusion and exclusion criteria that were not written in the article because when reviewing the literature in the presentation of the study subjects, they did not present these criteria. However, in the previous requirement it was clarified that all the participants had the same degree of training and had not previously participated in strength training programs.

Point 4. Details about the study participants are also relevant in terms of the reliability and validity of the strength assessments and the fact that a learning effects may have occurred between the series of tests.

Response 4.

The following is added:

All the participants underwent two weeks of adaptation with four sessions per week, performing four series of ten repetitions of each of the strength evaluation tests, with the purpose of standardizing the learning of the methodologies and protocols of each test.

Point 5. It is recommended, that subgroups of participants be explored based on training status

Response 5. There were no significant differences in the level of training of the participants; therefore, only one group was established.

Point 6. The Result section also needs minor adjustments. The Figures reported occupy large space of manuscript and a highly confusing. The key results should be reported in a more concise table with results for also reported in the text. The discussion focuses mainly on the results of the upper and lower body muscle strength test. The discussion section should also encompass aspects of dynamic athlete performance and how in the relationship is variables and how this investigation contributes to our understanding of the relationship in between strength and power in athletes.

Response 6.

A summary table was prepared that consolidates the information in a more dynamic way.

Variables

Gender

Variables with which it correlates

%

Significance level

Correlation Magnitude

W(n=96),M(n=259)       T (n=355)

P<0,05

p<0,01

Weak

Moderate

Strong

Very strong

CMJ (cm)

F

5

42%

20%

80%

80%

20%

0%

0%

M

10

83%

0%

100%

90%

10%

0%

0%

T

12

100%

0%

100%

33%

33%

33%

0%

DinamoDer

F

10

83%

10%

90%

70%

30%

0%

0%

M

8

67%

13%

88%

75%

25%

0%

0%

T

12

100%

0%

100%

33%

33%

33%

0%

PeakpowerWin

F

8

67%

50%

50%

100%

0%

0%

0%

M

10

83%

0%

100%

50%

25%

25%

25%

T

12

100%

0%

100%

25%

42%

33%

0%

Fuerza maxima Sentadilla(kg)

F

11

92%

9%

91%

73%

9%

18%

0%

M

8

67%

0%

100%

50%

25%

25%

0%

T

12

100%

0%

100%

33%

8%

42%

17%

Vel30mLanz

F

5

42%

20%

80%

60%

40%

0%

0%

M

7

58%

43%

57%

100%

0%

0%

0%

T

12

100%

0%

100%

33%

50%

17%

0%

Fuerza isometrica piernas

F

10

83%

30%

70%

80%

20%

0%

0%

M

8

67%

0%

100%

38%

63%

0%

0%

T

12

100%

0%

100%

33%

17%

50%

0%

FzaMaxPecho

F

9

75%

11%

89%

44%

33%

22%

0%

M

8

67%

13%

88%

38%

38%

25%

0%

T

12

100%

0%

100%

33%

0%

58%

8%

ValorPicoPotenciaPecho

F

10

83%

30%

70%

70%

20%

10%

0%

M

11

92%

18%

82%

64%

27%

9%

0%

T

12

100%

0%

100%

25%

25%

50%

0%

ValorPicoPotenciaSentadilla

F

11

92%

9%

91%

55%

36%

9%

0%

M

11

92%

18%

82%

55%

36%

9%

0%

T

12

100%

0%

100%

33%

17%

42%

8%

VMP50PECHO

F

7

58%

57%

43%

86%

0%

14%

0%

M

7

58%

14%

86%

86%

14%

14%

0%

T

12

100%

0%

100%

67%

25%

8%

0%

VMP70PECHO

F

6

50%

33%

67%

83%

0%

17%

17%

M

5

42%

0%

100%

80%

20%

20%

0%

T

12

100%

0%

100%

92%

0%

8%

0%

VMP50SENT

F

9

75%

11%

89%

900%

75%

11%

89%

M

6

50%

17%

83%

83%

0%

17%

0%

T

12

100%

0%

100%

83%

8%

8%

0%

VMP70SENT

F

9

75%

33%

67%

89%

0%

11%

0%

M

7

58%

14%

86%

71%

0%

14%

0%

T

12

100%

0%

100%

92%

0%

8%

0%

Point 6 The discussion focuses primarily on the upper and lower body muscular strength test results.

Response 6

The following is added

The present study provides evidence on the strong relationship (p<0.01) between the maximum strength and power tests. As has been described throughout this chapter, there is divergence between various studies regarding the relationship between these two manifestations of force. The exact relationship between maximal strength (measured by 1RM) and power output is unclear. Stone indicates that maximal strength is the basic quality that affects power output. It was initially established that maximal force affects power in a hierarchical manner with decreasing influence as external load decreases, up to a point where other factors, such as rate of force development, become more important. This argument indicates that maximal force should have its greatest influence on power output at heavy loads and that light loads should be less influenced by maximal force levels. However, some evidence indicates that maximal force may have a greater influence on power output over a broader range of loads than Moss et al. investigated the relationship between 1RM and PP at various percentages of 1RM using elbow flexion and identified strong correlations between 1RM and maximal PP output (r < 0.93), as well as a strong correlation between PP output at low % of 1RM (r < 0.73). In the present study, a strong correlation (r=0.81) was observed between maximal strength and PP in both the squat and chest press.